



# An Unusual Winter Ozone Event in Colorado

Andrew O. Langford[1], Raul J. Alvarez II[1], Kenneth C. Aikin[1,2], Sunil Baidar[1,2], W. Alan Brewer[1], Steven S. Brown[1], Matthew M. Coggon[1,2], Patrick D. Cullis[2,3], Jessica B. Gilman[1], Georgios I. Gkatzelis[1,2*], Detlev Helmig[4], Bryan J. Johnson[3], K. Emma Knowland[5,6], Rajesh Kumar[7], Aaron D. Lamplugh[1,2**], Audra McClure-Begley[2,3], Brandi J. McCarty[1,2], Ann M. Middlebrook[1], Gabriele Pfister[8], Jeff Peischl[1,2,3], Irina Petropavlovskikh[2,3], Pamela S. Rickly[1,2**], Andrew W. Rollins[1], Scott P. Sandberg[1], Christoph J. Senff[1,2], Carsten Warneke[1]

[1]NOAA Chemical Sciences Laboratory, Boulder, CO 80305, USA.

[2]Cooperative Institute for Research in Environmental Sciences, University of Colorado, Boulder, CO, 80309, USA.

[3]NOAA Global Monitoring Laboratory, Boulder, CO 80305, USA.

[4]Boulder Air LLC, Boulder, Colorado, 80301, USA.

[5]Morgan State University, GESTAR-II, Baltimore, MD, 21251, USA.

[6]NASA Goddard Space Flight Center, Global Modeling and Assimilation Office, Greenbelt, MD, USA.

[7]NSF NCAR Research Applications Laboratory, Boulder, CO, USA, 80307, USA.

[8]NSF NCAR Atmospheric Chemistry Observations and Modeling Laboratory, Boulder, CO, USA, 80307, USA.

*Now at Institute of Energy and Climate Research, IEK-8: Troposphere, Forschungszentrum Jülich GmbH, Jülich, Germany.

** Now at Air Pollution Control Division, Colorado Department of Public Health and Environment, Denver, CO, 80246, USA.

*Correspondence to: Andrew O. Langford (andrew.o.langford@noaa.gov)*





**Abstract.**

Surface ozone ($O_3$) mixing ratios exceeding the National Ambient Air Quality Standard (NAAQS) were measured at rural monitors along the Colorado Front Range on 17 April 2020 during the COVID-19 lockdown. This unusual episode followed back-to-back upslope snowstorms and coincided with the presence of a deep stratospheric intrusion, but ground-based lidar and ozonesonde measurements show that little, if any, of the $O_3$-rich lower stratospheric air reached the surface. Instead, the statically stable lower stratospheric air suppressed the growth of the convective

boundary layer and trapped nitrogen oxides ($NO_x = NO + NO_2$) and volatile organic compounds (VOCs) emitted by motor vehicles and oil and natural gas (O&NG) operations near the ground where the clear skies and extensive snow cover triggered a short-lived photochemical episode similar to those observed in the O&NG producing basins of northeastern Utah and southwestern Wyoming. In this study, we use a combination of lidar, ozonesonde, and surface measurements, together with the WRF-Chem and GEOS-CF models, to describe the stratospheric intrusion and

characterize the boundary layer structure, HYSPLIT back trajectories to show the low-level transport of $O_3$ and its precursors to the exceedance sites, and surface measurements of $NO_x$ and VOCs together with a 0-D box model to investigate the roles of urban and O&NG emissions and the COVID-19 quarantine in the $O_3$ production. The box model showed the $O_3$ production to be $NO_x$ saturated, such that the $NO_x$ reductions associated with COVID-19 exacerbated the event rather than mitigating it. Such winter $O_3$ exceedances may become more common in Denver

with expected, future $NO_x$ reductions.



## 1 Introduction

Ground-level ozone ($O_3$) is one of the six "criteria" pollutants deemed particularly harmful to human health and welfare and made subject to National Ambient Air Quality Standards (NAAQS) by the U.S. Clean Air Act (CAA) (Karstadt et al., 1993). Unlike the other gaseous criteria pollutants (*i.e.*, $NO_2$, $SO_2$, and CO), $O_3$ is not directly emitted by anthropogenic activities, but is a secondary pollutant formed by photochemical reactions of nitrogen oxides ($NO_x$ = NO + $NO_2$) and volatile organic compounds (VOCs) originating from natural or anthropogenic sources (Jaffe et al., 2018). Surface $O_3$ mixing ratios usually peak in summer when photochemical production is accelerated by warm temperatures and high solar fluxes (Sillman and Samson, 1995) and most exceedances of the NAAQS, currently set at 70 parts-per-billion by volume (ppbv) for the maximum daily 8-h average (MDA8), occur in or around urban areas during the so-called "ozone season" from May to September.

A striking exception to this paradigm was first described by *Schnell et al.* (2009) who documented multi-day wintertime episodes with hourly $O_3$ mixing ratios in excess of 150 ppbv in the rural Upper Green River Basin of Wyoming (Carter and Seinfeld, 2012; Rappenglück et al., 2014). Similar high-$O_3$ episodes were subsequently observed in the nearby Uinta Basin of northeastern Utah (Martin et al., 2011) and three Uintah Basin Winter Ozone Study (UBWOS) field campaigns (Ahmadov et al., 2015; Edwards et al., 2014; Edwards et al., 2013; Helmig et al., 2014) were conducted in 2012 - 2014 to investigate this phenomenon. These campaigns and related studies determined that the high $O_3$ episodes occurred when $NO_x$ and VOCs emitted by local oil and natural gas (O&NG) operations built up within the unusually shallow boundary layers that form above the basin-like terrain of these remote O&NG fields at very cold temperatures (Lareau et al., 2013; Oltmans et al., 2014; Neemann et al., 2015; Lyman and Tran, 2015; Schnell et al., 2016; Mccaffrey et al., 2019). Snow cover proved to be an essential component of these episodes that: i) enhanced the actinic flux by increasing the surface albedo, and ii) reinforced the boundary layer stability by decreasing the sensible heat flux (Bader and Mckee, 1985).

Persistent cold pools form in other O&NG fields with basin-like topography (Mansfield and Hall, 2018), but are much less likely to occur in those with more open terrain such as the Wattenberg Gas Field (WGF) in the Denver-Julesburg Basin of northeastern Colorado. The WGF, which encroaches on the northern Front Range urban corridor, lies within the Denver Metro/Northern Front Range (DM/NFR) $O_3$ non-attainment area, which was recently downgraded from "serious" to "severe 15" non-attainment of the 2008 $O_3$ NAAQS of 75 ppbv. Numerous studies (*e.g.*, (Pétron et al., 2012; Petron et al., 2014; Mcduffie et al., 2016; Cheadle et al., 2017; Flocke et al., 2020; Helmig, 2020)) have investigated the impact of O&NG emissions on $O_3$ mixing ratios in the DM/NFR, but there have been no reported instances of winter $O_3$ episodes like those seen in the Uinta and Green River Basins. Here, we describe such an event that resulted in the only NAAQS exceedances in the U.S. on 17 April 2020 (Fig. 1).

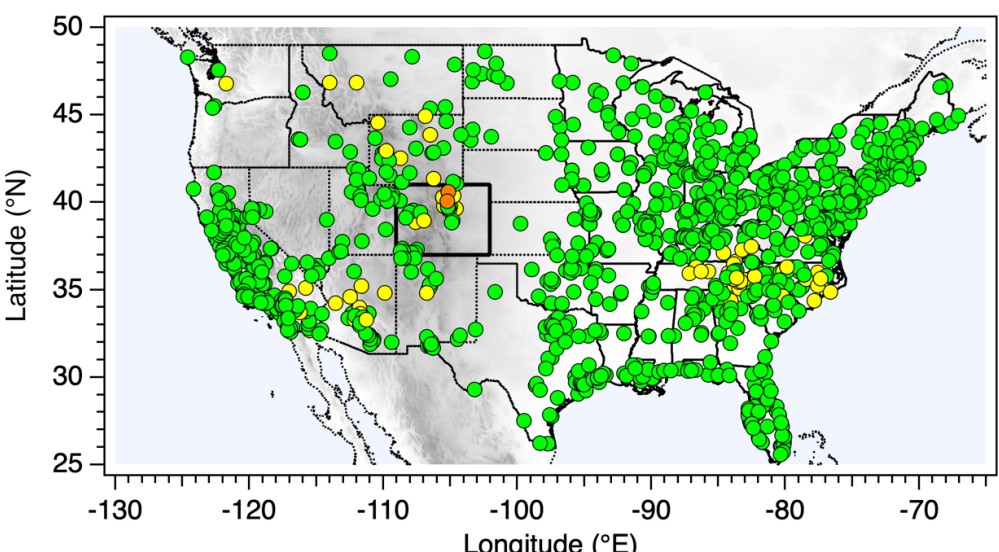

**Figure 1: MDA8 O₃ measured by the roughly 1200 regulatory monitors that reported to the U.S. EPA on 17 April 2020. Colorado is outlined in heavy black. The symbols are colored according to the EPA Air Quality Index (AQI) with green representing "good" (0-54 ppbv), yellow "moderate" (55-70 ppbv), orange "unhealthy for sensitive groups" (71-85 ppbv). The Boulder Reservoir (BOUR, 77 ppbv) and Ft. Collins-West (FTCW, 75 ppbv) monitors measured the highest MDA8 O₃ in the U.S. on that day.**

In this short-lived episode, which occurred early in the COVID-19 lockdown (He et al., 2023; Peischl et al., 2023), the shallow inversion was reinforced by the descent of statically stable lower stratospheric air deep into the lower troposphere. In this study, we use a combination of lidar, ozonesonde, and surface measurements, together with the WRF-Chem and GEOS-CF models, to describe the stratospheric intrusion and characterize the boundary layer structure, surface measurements of NO$_x$ and VOCs together with a 0-D box model (Rickly et al., 2023) to evaluate the production of O₃ by local photochemistry, and HYSPLIT back trajectories (Stein et al., 2015), together with VOC ratios, to assess the contributions of mobile sources and O&NG operations to the episode.

## 2 Meteorological Context

Back-to-back upslope storms associated with a slow-moving upper-level low brought heavy snow and record low temperatures to the DM/NFR between 11 and 16 April 2020. The winds shifted to the north as the first cold front arrived on the night of April 11 (Fig. 2a) and the temperature decreased from 15°C to -5°C in less than 6 hours with snow beginning in the early morning hours of 12 April. The snowfall was heaviest in and along the foothills west of Boulder where moist air from the Gulf of Mexico orographically lifted by the higher terrain collided with cold air from the north and west. Snow continued to fall through the next 48 hours with more than 42 cm (16.5 in) accumulating at the NOAA David Skaggs Research Center (DSRC) in southwest Boulder. Relatively little snow fell in the O&NG fields to the east, however, with only 3 cm of snow measured at the Denver International Airport (DIA) National Weather Service (NWS) office. The DIA station did measure a record low daily high temperature of -3.9°C on the





afternoon of 13 April, however, and the radiative cooling that occurred after the skies cleared later that night resulted in a record low temperature of -9.4°C on the morning of 14 April (Fig. 2b). The temperature climbed to 8°C by the afternoon, however, and the skies remained clear except for a few scattered clouds above the foothills.


The spring-like conditions were short lived, however, as the second storm arrived on the 15th and brought an additional 43 cm (16.9 in) of snow to the DSRC and another 61 cm (24 in) to the nearby foothills. This storm broke the seasonal snowfall record for Boulder which had stood since 1908-1909. This storm also brought significantly more snow to the eastern plains with 10 cm (4 in.) measured at DIA. The skies cleared around midnight and the subsequent radiative

cooling contributed to another record low of -11.7°C at the DIA on the morning of the 17th. As on the 14th, the skies remained clear throughout the day with the temperature peaking at 8°C in the afternoon.

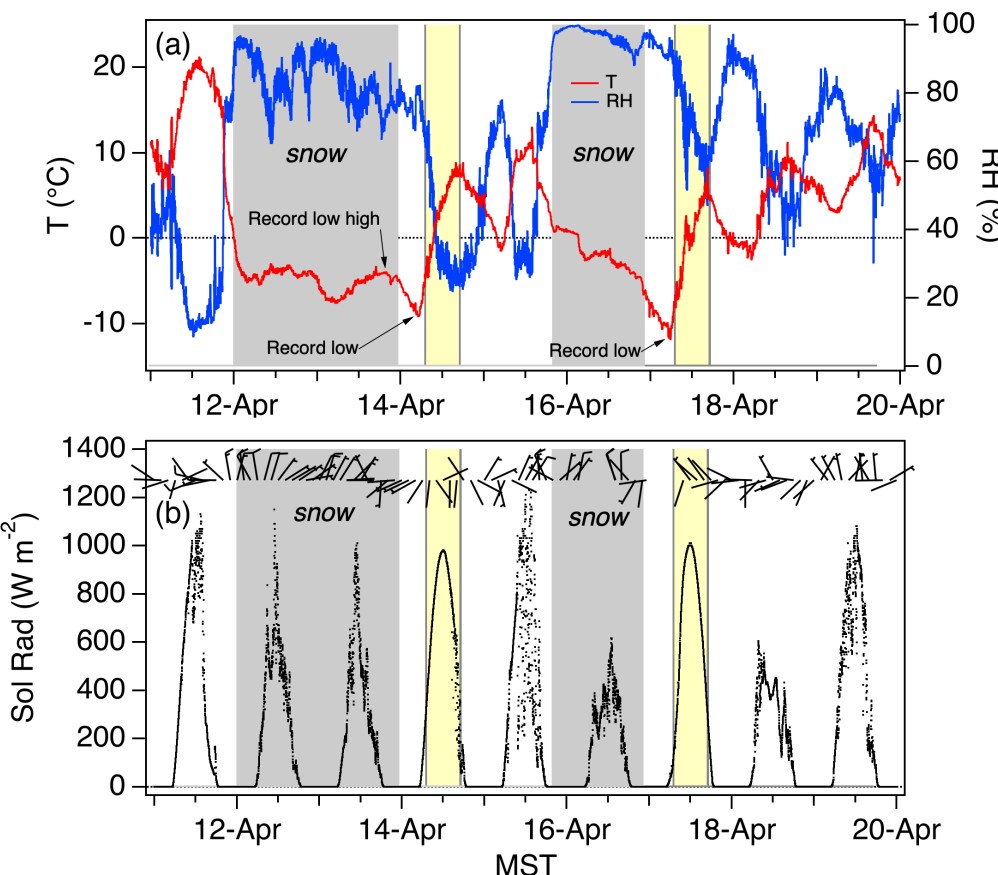

Figure 2: (a) Temperature and relative humidity, and (b) solar irradiance and wind measurements from the Longmont

Union Reservoir (LUR). Snow periods are highlighted in gray and the photochemically active periods on the 14th and 17th in yellow.



### 3 Surface Ozone

The similar temperatures and solar irradiance on the 14[th] and 17[th] suggest a similar potential for photochemical production, but Fig. 3 indicates that the measured $O_3$ mixing ratios on these two days were very different. Figs. 3a and 3b show the MDA8 $O_3$ mixing ratios measured by all of the regulatory and non-regulatory monitors operated by private, state, and federal agencies (see Supplement) on 14 and 17 April respectively, superimposed on the corresponding (cloud-free) *Terra*-MODIS images, which show more extensive snow cover across the WGF on the

17[th]. The filled circles represent the regulatory monitors from Fig. 1 and include the Boulder Reservoir (BOUR) and Ft. Collins-West (FTCW) monitors that exceeded the NAAQS with MDA8 mixing ratios of 77 and 75 ppbv, respectively on the 17[th]. The filled squares show non-regulatory research monitors operated by NOAA GML at Niwot Ridge (NWR) and Table Mountain (BOS), and by NOAA CSL at the DSRC in Boulder. The filled diamonds show the non-regulatory monitors operated by Boulder A.I.R. at the Longmont Municipal Airport (LMA) and Longmont Union

Reservoir (LUR). The measurements from these sites are described in more detail in the Supplement. Fig. 3a shows that all of the monitors in the DM/NFR measured "good" MDA8 $O_3$ mixing ratios on the 14[th], but Fig. 3b shows that most measured mixing ratios considered "moderate" or even "unhealthy for sensitive groups" on the 17[th]. The highest MDA8 $O_3$ was measured in the largely rural area bounded by the WGF and the foothills between Boulder and Ft. Collins; the lowest was measured in the more densely populated Denver urban area.


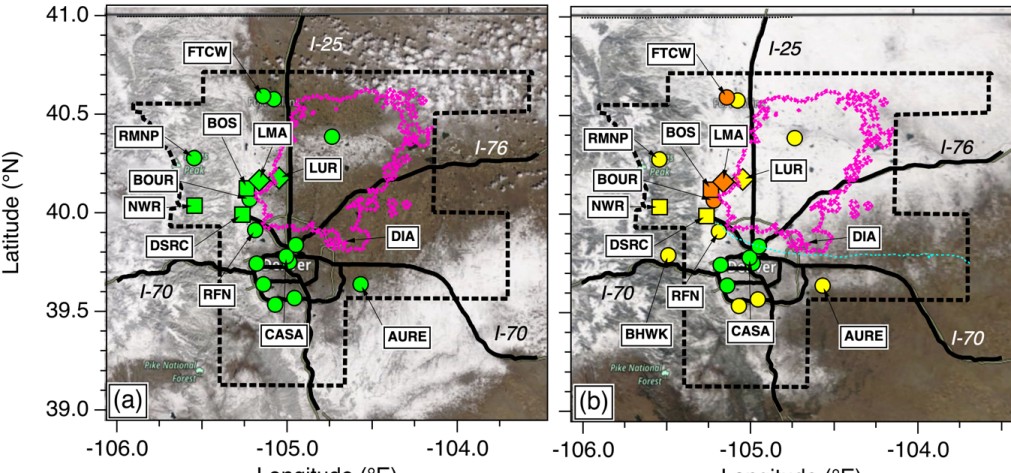

**Figure 3: MDA8 $O_3$ mixing ratios (AQI) on (a) 14 April and (b) 17 April 2020, superimposed on the daily (1030 local time) corrected reflectance (True Color) Terra/MODIS images from NASA/Worldview (https://worldview.earthdata.nasa.gov/, last accessed 24 June 2024). The black dashed lines show the DM/NFR $O_3$ non-attainment area and the dotted magenta lines the WGF. The dotted cyan line in (b) shows the path of the GML ozonesonde (Section 4.2).**





Fig. 4a displays time series of the hourly $O_3$ measurements from all of the monitors in Fig. 3. The measurements from the high elevation (>2.5 km above mean sea level, a.s.l.) "baseline" monitors at Blackhawk (BHWK), NWR, and Rocky Mountain National Park (RMNP) are plotted in black with the measurements from those monitors that recorded higher (lower) MDA8 $O_3$ on the $17^{th}$ plotted in red (blue). The heavy red and blue lines show the measurements from

the BOS and CASA monitors, which recorded the highest and lowest $O_3$ mixing ratios, respectively, on 17 April. Fig. 3b shows that the three baseline monitors are almost equally spaced (about 25 km) along a roughly north-to-south line in the mountains to the west of the NFR, and the measurements from these 3 sites appear very similar (the BHWK monitor was offline on 14 April), suggesting that the measurements were usually representative of the lower free troposphere.


       Fig. 4b is similar to Fig. 4a, but plots only the mean baseline mixing ratios (±5 ppbv) and the BOS (red) and CASA (blue) time series. The NOAA GML BOS monitor is located at the Department of Commerce field site on Table Mountain, a grass-covered mesa (1.69 km a.s.l.) about 15 km north of the DSRC and 6 km north of the BOUR monitor. The CASA monitor is located 33 km southeast of the DSRC at the La Casa NCORE site (1.60 km a.s.l.) near the

confluence of I-25 and I-70 in Denver. The BOS measurements were very similar to the baseline measurements between the $11^{th}$ and $16^{th}$, indicating that there was little photochemical production of $O_3$ or deposition to the snow-covered surface (Helmig et al., 2007) during this wintry period. The BOS monitor measured nearly 20 ppbv more $O_3$ than the baseline monitors on the afternoon of the $17^{th}$, and 10-15 ppbv more $O_3$ on the $18^{th}$. The CASA measurements appear very different, however, and were 5-10 ppbv lower than the baseline mixing ratios during the day and as much

as 50 ppbv lower during the late night and early morning. This shows net destruction of $O_3$ in this high traffic area, and the effects of $NO_x$ titration are particularly striking in the measurements from the morning of the $17^{th}$ when all of the surface $O_3$ was titrated between 0300 and 0500 MST. The dashed blue line shows that the total $O_x$ mixing ratios (*i.e.*, $NO_2 + O_3$) at CASA were accordingly similar to the baseline $O_3$ between the $11^{th}$ and $16^{th}$. The $O_x$ was higher than the baseline $O_3$ on the $17^{th}$ and $18^{th}$, yet nearly 10 ppbv smaller than the average $O_3$ measured by the rural BOS

monitor to the northwest the following night.





**Figure 4: (a)** Time series of the hourly O$_3$ measurements from all of the monitors shown in Fig. 3. The measurements from the high elevation baseline sites (BHWK, NWR, and RMNP) are plotted in black. The measurements from the BOS and CASA monitors that measured the highest and lowest O$_3$ on 17 April are plotted in red and blue, respectively. **(b)** Same as (a), but with only the measurements from the BOS (red) and CASA (blue) monitors together with the mean baseline mixing ratios (black, ±5 ppbv). The dashed blue line shows the CASA O$_x$ (NO$_2$ + O$_3$) mixing ratios and the horizontal dashed lines the 2015 NAAQS.





## 4 Stratospheric Influence

### 4.1 Upper air analyses

It is well established that deep stratospheric intrusions can increase surface $O_3$ and even cause exceedances of the
NAAQS in the NFR (Langford et al., 2009) and the upper-level low responsible for the late season snowstorms also
spawned several intrusions as it crossed North America. The 300 hPa (≈9.2 km a.s.l.) potential vorticity (PV)
distributions, a tracer for lower stratospheric air, from the NASA MERRA-2 Reanalysis (Gelaro et al., 2017) displayed
on the left side of Fig. 5 show the progression of two of these intrusions across the southern U.S. and the corresponding
5 km a.s.l. (≈540 hPa) $O_3$ distributions from the NCAR Weather Research Forecasting with Chemistry (WRF-Chem)
(Kumar et al., 2021) model on the right show their impacts on mid-tropospheric $O_3$.

Fig. 5a shows an elongated trough trailing behind the main low-pressure center over eastern Canada and curving
cyclonically across Colorado and the Southwestern U.S. on the afternoon of 14 April. The band of elevated PV along
the leading edge of the trough near the southeast corner of Colorado shows the tropopause fold, but Fig. 5b indicates
that the equatorward sloping tongue of high $O_3$ passed well to the south of Colorado. The intrusion moved eastward
and into the Atlantic Ocean over the next couple of days as a second trough dropped down from Canada (Figs. 5c and
5e) and developed a tropopause fold above the Pacific Northwest (Fig. 5d). This intrusion evolved into an elongated
streamer as it moved into Colorado (Fig. 5f) behind the snowstorm. The PV analysis for the afternoon of 17 April in
Fig. 5g shows the narrow filament of high PV air stretching across southern Wyoming and northeastern Colorado and
the WRF-Chem analysis (Fig. 5h) shows the narrow filament of high $O_3$ directly above Colorado.

Fig. 6 displays vertical cross-sections of the $O_3$ distributions from the NASA Goddard Earth Observing System
composition forecast (GEOS-CF) model (Keller et al., 2021; Knowland et al., 2022) along latitudinal and longitudinal
transects through Boulder (vertical dashed lines) on the afternoons of 14 and 17 April. Fig. 6a and 6b show the
downward sloping tongue of lower stratospheric air from the first intrusion far descending isentropically well to the
south of Boulder in agreement with Figs. 5a and 5b, and Fig. 6c shows the second intrusion poised directly above
Colorado in agreement with Figs. 5g and 5h. Figs. 6c and 6d show this deep intrusion increasing the surface mixing
ratios to 50-60 ppbv along the foothills near Boulder and to 60-65 ppbv in the mountains to the south, in agreement
with the measurements from the high elevation baseline monitors (cf. Fig. 4b).



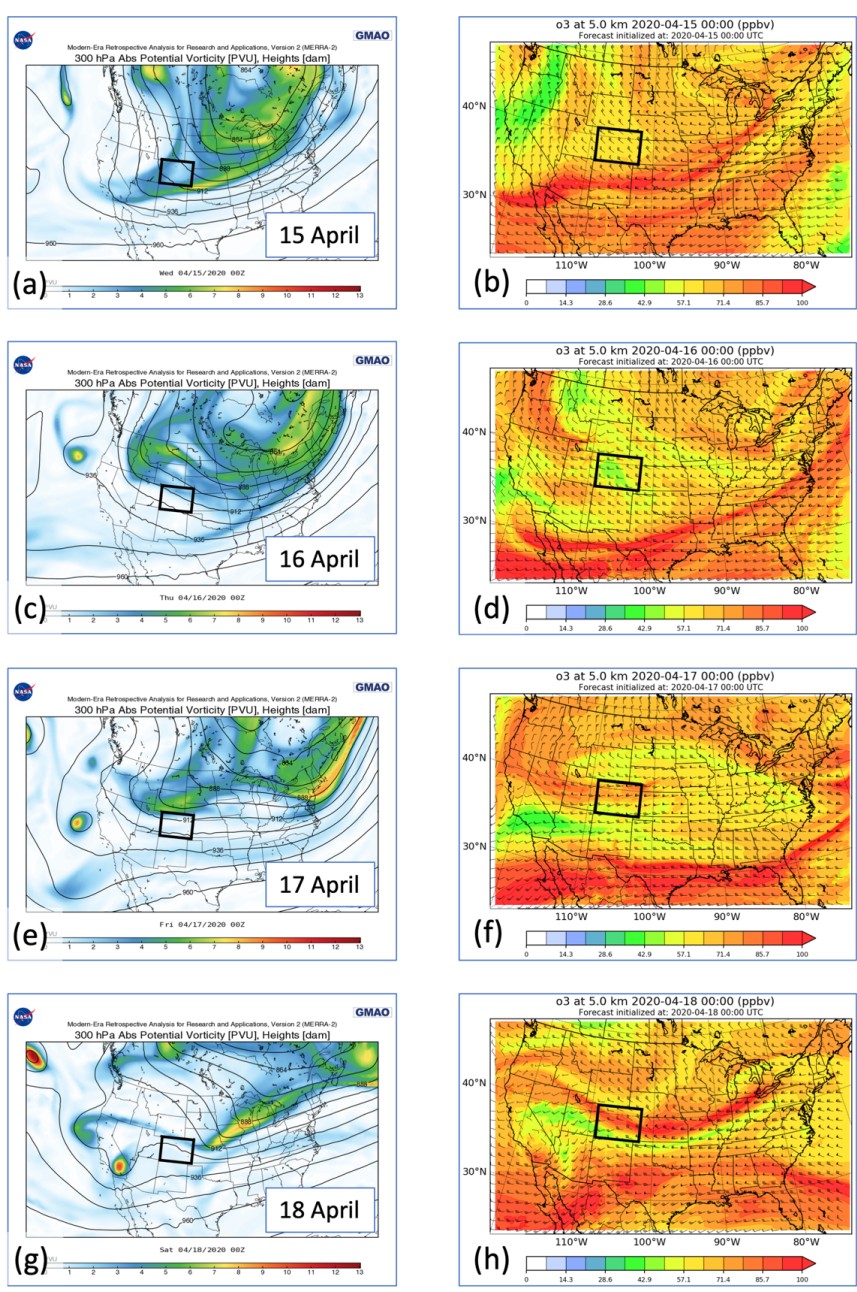


**Figure 5: NASA MERRA-2 300 hPa PV distributions (left) and NCAR WRF-Chem 5 km a.s.l. O₃ distributions (right) at, from top to bottom, 0000 UT on 15, 16, 17, and 18 April (1700 MST on 14, 15, 16, and 17 April). Colorado is outlined in black. The images were downloaded from the publicly accessible archives at**

**https://fluid.nccs.nasa.gov/reanalysis/classic_merra2/ (last access 24 June 2024) as in Duncan et al. (2021) and https://www.acom.ucar.edu/firex-aq/forecast.shtml (last access 24 June 2024).**



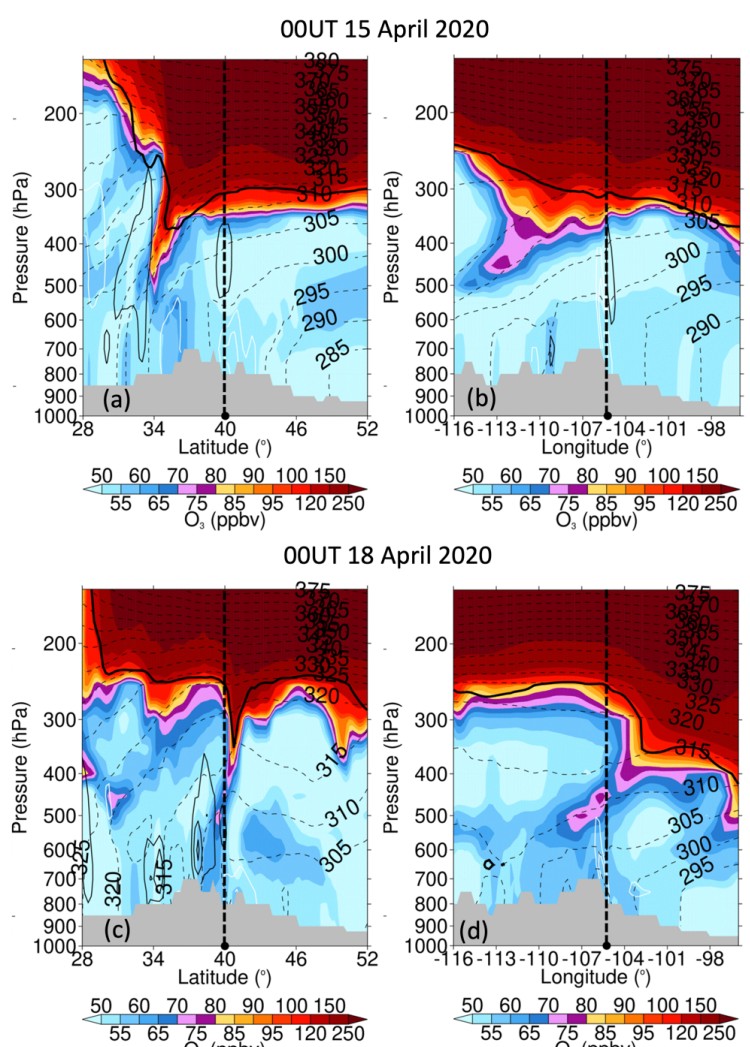


**Figure 6: Vertical cross-sections of the O₃ mixing ratios (color) along latitudinal (left) and longitudinal (right) transects passing through Boulder (vertical dashed lines) at 0000 UT on April 15 (top) and April 18, 2020 (bottom) from the NASA GEOS-CF model. Equivalent potential temperature (dashed contour lines, 5 K intervals) and the dynamical tropopause isosurface of 2 potential vorticity units (PVU: $10^{-6}$ K m² kg⁻¹ s⁻¹; thick black contour), vertical velocity (solid contour lines, 10 hPa h⁻¹ intervals, with white contours for descent and black contours for ascent) are drawn (Knowland et al., 2017). Orography indicated by grey region.**


### 4.2 Upper air measurements

The GEOS-CF analyses can also be compared to the O₃ vertical distributions measured by the ground-based TOPAZ

lidar at the DSRC (Alvarez et al., 2011; Langford et al., 2019). TOPAZ, which is part of the NASA-supported



Tropospheric Ozone Lidar Network (TOLNet) (Newchurch et al., 2016), measures $O_3$ and 0.294 μm aerosol backscatter (ß) profiles from about 20 m above ground level (a.g.l.) to roughly 8 km a.g.l. (depending on the total extinction and solar background) with a time resolution of 10 min and with an effective $O_3$ vertical resolution ranging from roughly 10 m near the surface to about 900 m at the far end of the measurement range. The total uncertainty of the $O_3$ measurements increases from approximately 3 ppbv below 4 km a.g.l. to about 10 ppbv at 8 km a.g.l. NOAA CSL also operated a commercial Doppler lidar that measured aerosol backscatter in the near-infrared (1.54 μm), horizontal windspeed and direction, and vertical velocity variance from the roof of the DSRC.

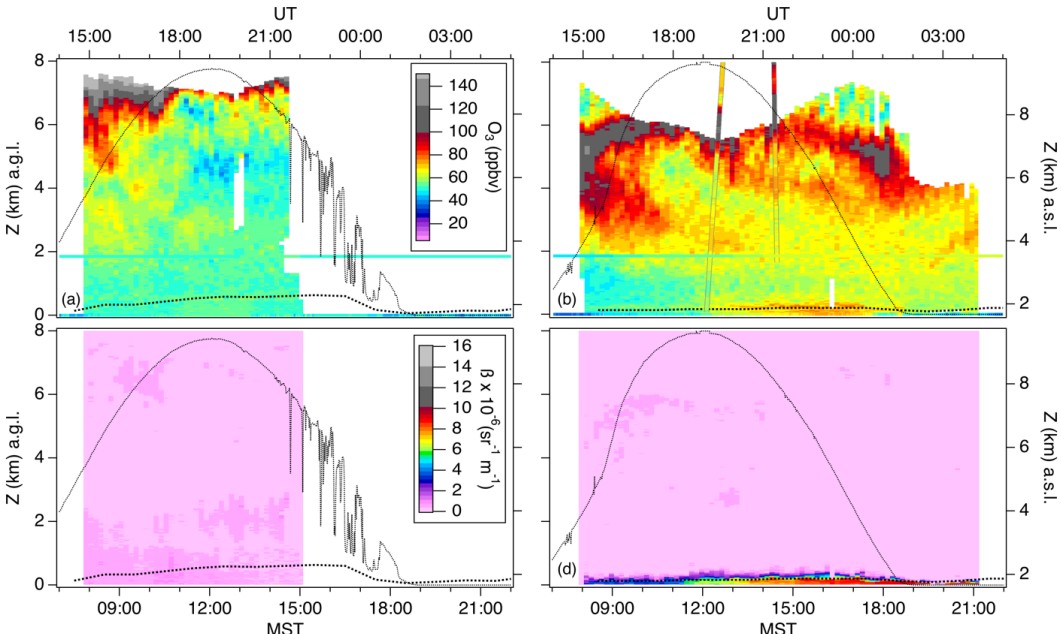

**Figure 7: Time-height curtain plots of the $O_3$ (top) and 0.294 μm backscatter (ß, bottom) distributions measured by TOPAZ on 14 April (left), and 17 April (right). The co-located *in-situ* $O_3$ measurements are plotted along the bottoms of the upper curtains and the ascending and descending $O_3$ profiles from the Marshall ozonesonde are superimposed on (b). The black traces show the normalized LUR solar irradiance and the dotted curves near the bottom the boundary layer heights inferred from the DSRC Doppler lidar measurements.**

Fig. 7 displays time-height curtain plots of the $O_3$ (top) and backscatter (bottom) distributions measured by TOPAZ on 14 April (left) and 17 April (right). The colored horizontal stripes in the upper panels show the *in-situ* measurements from the monitor located in the TOPAZ truck, which sampled air at 5 m a.g.l., and the NWR monitor located 28 km NNW at an elevation of 3.52 km a.s.l. (1.85 km higher than TOPAZ). The sloping lines in Fig. 7b show the ascending and descending profiles from an ozonesonde launched by NOAA GML from the NCAR Marshall Field Site (39.949°N, -105.197, 1.74 km a.s.l.) about 8 km SE of the DSRC at 1207 MST. The strong westerlies aloft carried the ozonesonde due east (cf. Fig. 3b) and the descending profile was measured about 140 km downwind (ESE) of the lidar. The dotted lines near the bottom show the planetary boundary layer (PBL) heights estimated from the co-located Doppler lidar





measurements of backscatter, wind shear, and turbulence (Choukulkar et al., 2017; Bonin et al., 2017). The PBL height peaked at 630 m on the 14[th], but did not exceed 200 m on the 17[th].

The TOPAZ $O_3$ measurements from 14 April (Fig 7a) show high (>80 ppbv) mixing ratios in the upper troposphere during the morning, but only moderate (40-60 ppbv) mixing ratios later in the day. These mixing ratios and the low

tropopause are both consistent with the GEOS-CF analyses in Fig. 6. The lidar measurements from 17 April (Fig 7b) show much higher mixing ratios throughout the mid-troposphere with high (>70 ppbv) $O_3$ measured as low as 3 km a.s.l. The lidar measurements from the afternoon show mixing ratios in excess of 60 ppbv at 2 km a.s.l. that are consistent with both GEOS-CF and the *in-situ* measurements from the three baseline sites (Fig. 4b). However, Fig. 7b shows a gap between the $O_3$ aloft and even higher mixing ratios near the surface. The backscatter measurements in

Fig. 7d also show much higher aerosol near the surface than was measured in the free troposphere on either day.

### 4.3 Boundary layer measurements

Figs. 8a and 8c show the hourly-averaged horizontal winds from the Doppler lidar superimposed on the lowest 1 km from the $O_3$ and ß curtains in Figs. 7b and 7d. The plots on the right show the potential temperature (theta) relative

humidity (RH), $O_3$, and wind profiles from the Marshall ozonesonde, and vertical velocity variance profiles from the Doppler lidar. Fig. 8a shows that the $O_3$ mixing ratios at 1 km were similar to the baseline measurements (≈55-60 ppbv) in the morning, but the $O_3$ and ß near the surface began to increase around 1300 MST, and grew rapidly after the northeasterly flow became established in the afternoon. The $O_3$ and aerosol both decreased after sunset when the low-level winds rotated to the south, but partially recovered when the winds shifted back to the northwest around 2100

MST. The midday sounding (Figs. 8b and 8d) shows a rapid decrease in relative humidity within an unusually strong temperature inversion extending from the surface to nearly 400 m. The mean-squared Brunt-Väisälä frequency of $N^2$ = 3.7 x $10^{-4}$ $s^{-2}$ within this inversion was comparable to that of the stratosphere (U.S., 1976).

The balloon sounding and the daytime Doppler lidar measurements both show westerly winds above 400 m and

easterly to northeasterly winds below 200 m. The region between 200 and 400 m delineated by the dashed lines and gray bands in Fig. 8 was characterized by southeasterly to southerly winds. The afternoon Doppler lidar measurements show that there was strong shear-induced turbulent mixing between the top of this transition layer and the lower free troposphere, but very little mixing between the PBL and the bottom of the transition layer. This is consistent with Fig. 8c, which shows the highest backscatter was confined to the PBL, and suggests that the elevated $O_3$ in the transition

layer (Fig. 8a) was advected to Boulder from the south.



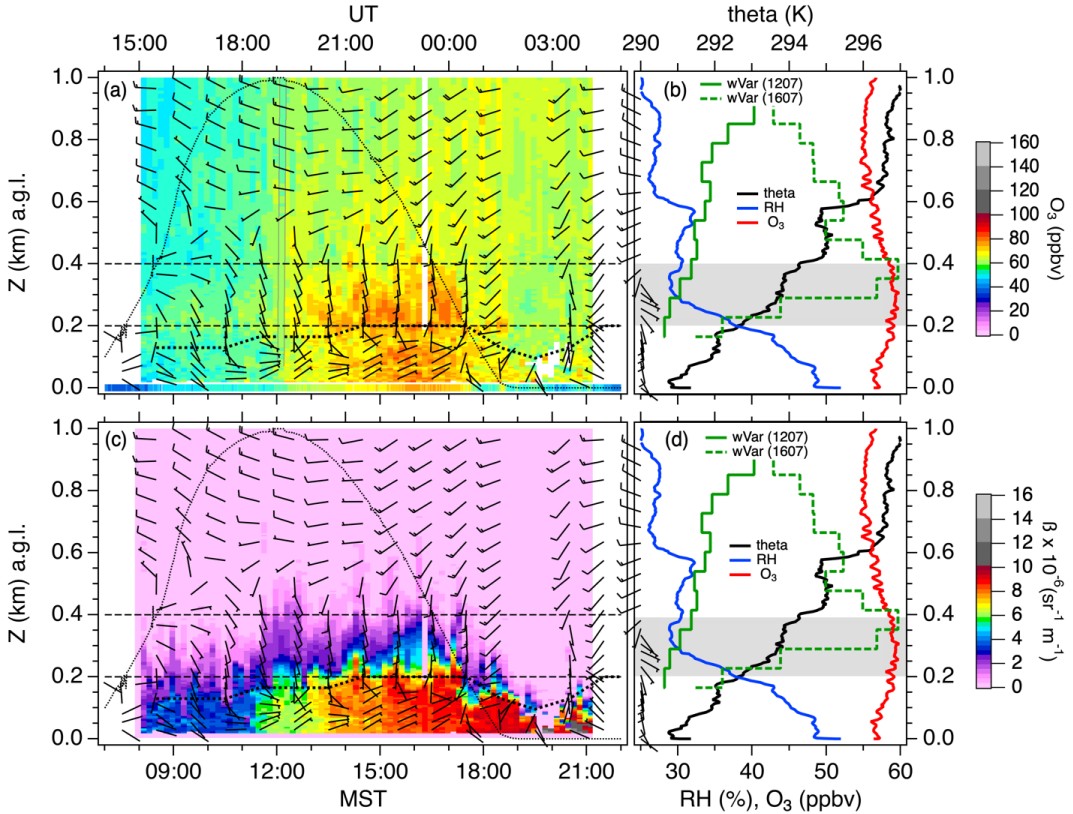

**Figure 8: (left) Expanded views of the (top) O$_3$ and (bottom) ß curtain plots from 17 April. The bar along the bottom of the**
**O$_3$ curtain shows the *in-situ* measurements from the DSRC. The barbs show the hourly-averaged winds and the heavy**
**dotted curves the PBL heights from the DSRC Doppler lidar. The black dotted curves show the normalized solar flux. (right)**
**Potential temperature, relative humidity, and wind profiles from the ascending ozonesonde launched at 1207 MST. The**
**solid and dashed green traces show the vertical velocity variance profiles (wVar) from the 1207 and 1607 MST (dashed)**
**Doppler lidar measurements (the scale ranges from 0 to 1 m$^2$ s$^{-2}$). The horizontal dashed lines in the curtain plots and gray**
**region in the ozonesonde plots are described in Section 4.3.**

## 5 Low-level transport

The rapid changes in near surface O$_3$ and ß created by shifts in wind direction (Fig. 8) show that low-level transport
had a large influence on the 17 April measurements. This is shown more explicitly in Fig. 9 which displays 24-h
HYSPLIT back trajectories (https://www.ready.noaa.gov/HYSPLIT_traj.php, last access 24 June 2024) (Stein et al.,
2015) launched 50 m above each of the O$_3$ monitors at 1600 MST (2300 UT) on 14 April (left) and 17 April (right).
The trajectories were calculated using meteorology from the National Centers for Environmental Prediction (NCEP)
3 km High-Resolution Rapid Refresh (HRRR) model (Benjamin et al., 2016). The top panels show the trajectory
horizontal paths and the bottom panels the corresponding altitudes. The trajectories from all of the monitors to the
south of Rocky Flats-North (RFN) are plotted in green.





The plots on the left (Figs. 9a and 9c) show that nearly all of the trajectories approached the DM/NFR from the west
on 14 April and descended rapidly from the lower free troposphere over the Rocky Mountains to the surface. Most of
these fast-moving trajectories were more than 800 km to the northwest over southwestern Montana 12 hours earlier
(not shown) with the two Ft. Collins trajectories following a lower and more southerly route that passed over Idaho.
The plots on the right (Figs. 9b and 9d) show that most of the trajectories launched from the monitors to the south of
the DSRC on the afternoon of 17 April also descended from the lower free troposphere above the Rocky Mountains,
but passed over the Denver metropolitan area before they reached the surface. The parcels doubled back when they
encountered the shallow southeasterly upslope flow in the boundary layer, however, and passed slowly through the
urban area near the surface. The only exception was the trajectory launched from the rural Aurora East (AURE)
monitor, which descended directly from the free troposphere. Even the trajectory launched from RFN monitor, which
is only 11 km from Boulder, but lies about 150 m higher and over a slight ridge, arrived from the southeast. Fig. 9b
shows that several of these trajectories passed over the Boulder area as they descended, but were more than 2 km
above the surface at the time.

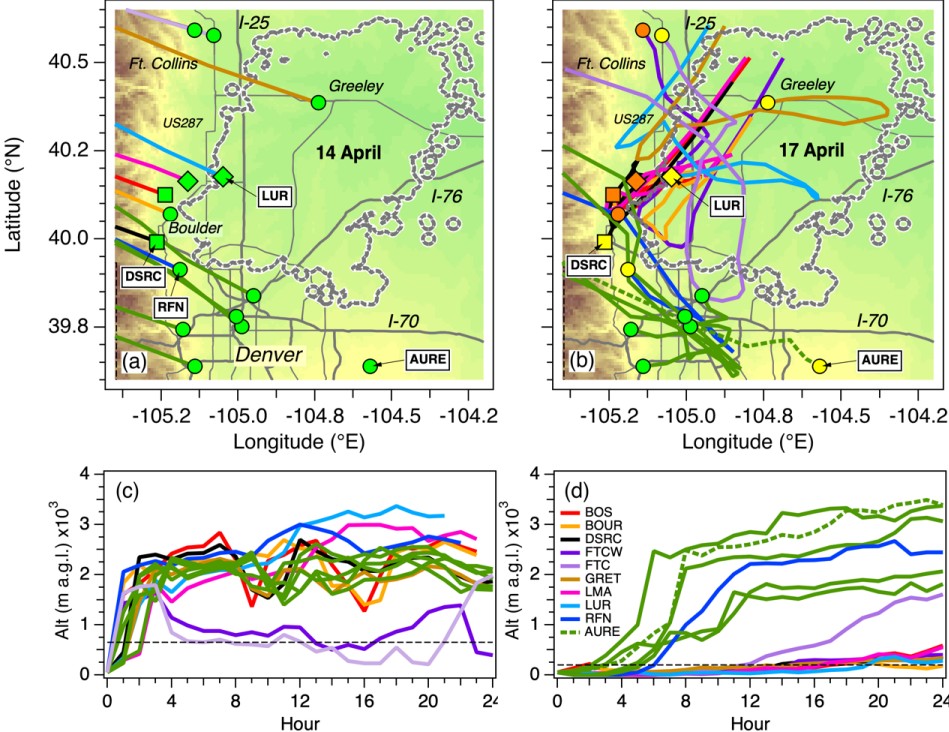

Figure 9: (top) Expanded views of the Northern Front Range showing the MDA8 O$_3$ measurements (AQI scale) and 24-h
HYSPLIT back trajectories initialized at 16 MST (23 UT) on 14 April (left) and 17 April (right). The trajectories launched
from all of the monitors to the south of Rocky Flats are shown in green. The dark gray lines show the major interstate (thick
solid lines) and US highways (thin solid lines). The WGF is outlined in dotted gray. (bottom) Altitude profiles of the 24-h
back trajectories. The dashed line shows the 200 m PBL height.






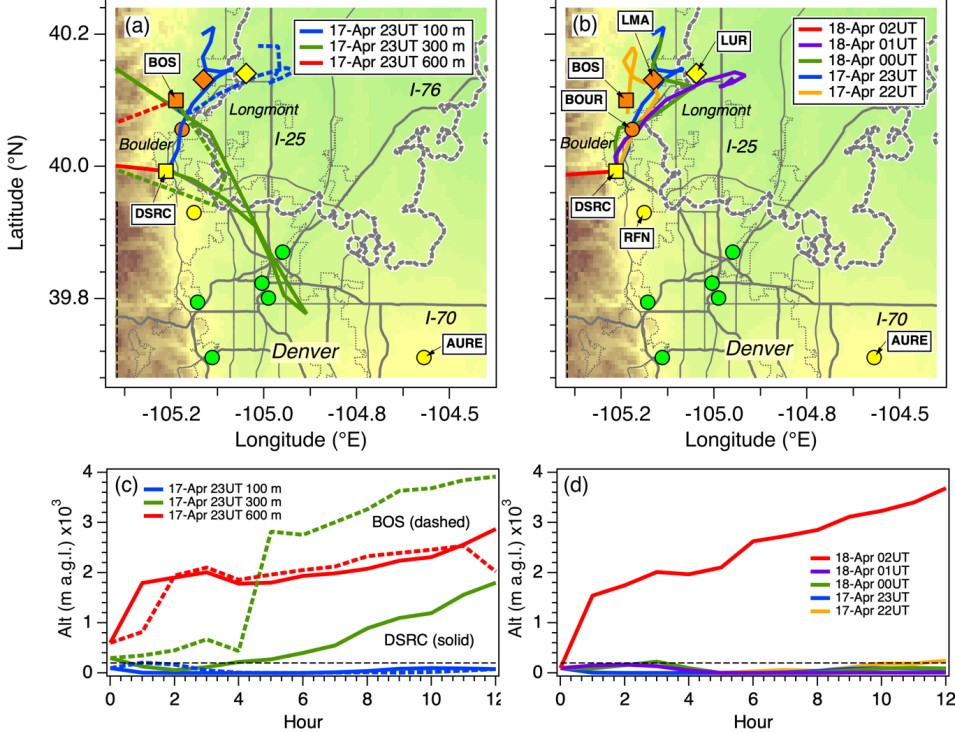


**Figure 10: (a) Relief map of the Boulder-Longmont area showing the MDA8 O₃ measurements and 12-h HYSPLIT back trajectories launched 100, 300, and 600 m above the DSRC at 1600 MST (2300 UT) on 17 April. (b) Same as (a), but showing the back trajectories launched 100 m above the DSRC every hour from 15 to 18 MST (22 to 01 UT) on the afternoon of 17 April. (c) Altitudes of the trajectories shown in (a). (d) Altitudes of the trajectories shown in (b). The dashed line shows the 200 m PBL height.**


Fig. 10a shows 12-h backward trajectories launched 100, 300, and 600 m above the DSRC and BOS monitors at 1600 MST (2300 UT) on 17 April. The DSRC trajectories came from different directions as would be expected from the Doppler lidar winds in Fig. 8. The trajectory launched in the lower free troposphere (600 m, red) descended rapidly

from the Rocky Mountains to the west, but the trajectory launched in the PBL (100 m, blue), approached the DSRC from the northeast after slowly meandering through the region with the highest O₃ mixing ratios. The 100 and 600 m BOS back trajectories followed similar paths. Both of the intermediate trajectories (300 m, green) approached from the southeast, with the BOS trajectory circling back to the west more than 500 m above Boulder. The DSRC trajectory appears similar to the RFN trajectory in Fig. 9b, however, which suggests that the elevated O₃ between 200 and 400

m above the DSRC in Fig. 8a may have originated, at least in part, from Denver.





Fig. 10b shows a series of back trajectories launched from the DSRC at hourly intervals between 2200 and 0200 UT (1500 to 1900 MST) on the 17[th]. The first 4 trajectories arrived from the northeast below 250 m, circling past the monitors with the highest $O_3$ passing with the 1600-1800 MST back trajectories passing close to the LUR monitor and I-25, the primary north-south route between Denver and Cheyenne, Wyoming and a major $NO_x$ source, between 5 and 6 hours earlier. The last trajectory (1900 MST or 0200 UT) descended from the west and the lower free troposphere.

## 6 Photochemical production

The clear skies, snow-covered ground, weak winds, and shallow boundary layer on 17 April 2020 provided the necessary meteorological conditions for a winter ozone event like those in the Uinta Basin, and the trajectory analyses in Section 5 show that the highest $O_3$ mixing ratios were measured in air that had lingered over the western half of the snow-covered WGF earlier in the day. The air sampled by these monitors had thus been exposed to $NO_x$ and VOCs emitted by traffic along I-25 in addition to VOCs from O&NG sources. Motor vehicle exhaust has high concentrations of ethyne as well as the reactive alkenes (e.g., ethene and propene) and aromatics (e.g., benzene and toluene) that contribute to $O_3$ production in urban areas (Nelson and Quigley, 1984; Broderick and Marnane, 2002; Borbon et al., 2013; Gentner et al., 2013; Thompson et al., 2015; Abeleira et al., 2017).

### 6.1 $NO_x$ and VOC measurements

Boulder A.I.R. also measures $NO_x$, VOCs, and other parameters at the LUR and BOUR monitoring sites (Pollack et al., 2021) (Supplement), and the HYSPLIT trajectories in Fig. 10 imply that the LUR measurements from the morning of 17 April should be representative of the air sampled by the BOS, BOUR, LMA, and DSRC monitors later that day. NOAA CSL also measured $NO_x$ and VOCs in mid-April as part of an extensive suite of *in-situ* surface measurements conducted during the COVID-AQS study (Peischl et al., 2023; Rickly et al., 2023). Fig. 11 compares the (a) $O_3$, (b) $NO_x$, (c) carbon monoxide (CO) or carbon dioxide ($CO_2$), and (d) $PM_{2.5}$ measurements from the LUR, BOUR, and DSRC made between 14 and 19 April. The second snowstorm disrupted the BOUR measurements on 16-17 April so Fig. 11a also plots the $O_3$ measurements from the nearby BOS monitor. The DSRC measurements were also disrupted, but for only a few hours. The mean baseline $O_3$ from Fig. 4b is also plotted, for reference. The DSRC and A.I.R. measurements are described in more detail in the Supplement.

Fig. 12 is similar to Fig. 11, but plots the corresponding (a) propane ($C_3H_8$), (b) *n*-butane (*n*-$C_4H_{10}$), (c) ethene ($C_2H_4$), and (d) ethyne/acetylene ($C_2H_2$) measurements. Note that the DSRC propane and *n*-butane measurements have been scaled up by a factor of 4 to better show the trends. Propane and *n*-butane originate primarily from O&NG activities (Gilman et al., 2013) while ethene and ethyne are major components (approximately 10% by weight) of motor vehicle exhaust (Nelson and Quigley, 1984; Broderick and Marnane, 2002; Gentner et al., 2013; Thompson et al., 2015). As in previous studies (*e.g.*, Gilman et al., 2013), we use propane and ethyne as tracers for O&NG and motor vehicle (or other combustion) influences, respectively, in our analysis.







**Figure 11: Time series of (a) O₃, (b) NOₓ (c), CO (DSRC) and CO₂ (LUR), and (d) PM₂.₅ measurements from the LUR (red), BOUR (green), and DSRC (blue) stations. The mean baseline O₃ (±5 ppbv) and BOS (dashed green) measurements are also plotted in (a). Panel (d) shows both the particle number density measured by the ultra-high sensitivity aerosol spectrometer (UHSAS) at the DSRC and the mass concentrations from the LUR station and the regulatory monitor in downtown Boulder. Snow periods are highlighted in gray and the photochemically active periods on the 14th and 17th in yellow.**





Figure 12: Same as Fig. 11, but showing the corresponding: (a) C$_3$H$_8$, (b) $n$-C$_4$H$_{10}$, (c) C$_2$H$_4$, and (d) C$_2$H$_2$ measurements. The DSRC C$_3$H$_8$ and $n$-C$_4$H$_{10}$ measurements have been scaled up by a factor of 4. Snow periods are highlighted in gray and the photochemically active periods on the 14$^{th}$ and 17$^{th}$ in yellow.

405

410



The $O_3$ time series in Fig. 11a shows that the measurements from the LUR, BOS, BOUR, and DSRC monitors differed from the baseline mixing ratios by less than ±5 ppbv on the afternoon of the 14th when the HYSPLIT trajectories (Fig. 9a) indicate that they all would have sampled free tropospheric air. The peak $O_3$ mixing ratios differed by as much as 20 ppbv on the afternoon of the 17th, however, with the nearly identical DSRC and LUR measurements lying halfway between the highest $O_3$ mixing ratios measured by the BOS and BOUR monitors and the baseline mixing ratios from the high elevation sites.

In contrast to $O_3$, Fig. 11b shows that the $NO_x$ mixing ratios were much higher at the LUR station than at the DSRC on the 17th. The LUR measurements are dominated by large peaks at 0145 and 0730 MST that are also seen in the $CO_2$ (Fig. 11c), ethene (Fig. 12c), and ethyne (Fig. 12d) time series, and are mirrored by large decreases in the $O_3$ time series. The peak $NO_x$ (79 ppbv) and $CO_2$ (522 ppmv) mixing ratios were the highest recorded during the entire month of April and the timing of the second peak is consistent with morning rush hour traffic. The DSRC measurements show smaller peaks around 0830 MST consistent with this interpretation. The LUR measurements also show smaller peaks on the mornings of the 14th, 15th, and 18th which confirms that the unusually high concentrations on the 17th were caused by the exceptionally shallow inversion layer which trapped more of the emissions near the surface. The LUR monitoring station is located 6 km west of I-25, the primary north-south route between Denver and Cheyenne. Figs. 11 and 12 show that the motor vehicle emissions decreased when the winds shifted to the northwest at sunrise, and the concentrations of propane (Fig. 12a) and *n*-butane (Fig. 12b) increased. This may reflect the influence of a nearby (0.3 km) O&NG well that was active in April 2020, but has since been decommissioned.

The scatter plots in Fig. 13 compare the LUR *n*-butane, propane, ethene, and ethyne measurements from 16-18 April 2020 with the relationships found in earlier WGF studies by *Gilman et al.* (2013) and *Pollack et al.* (2021). The solid lines show the orthogonal distance regression (ODR) fits to the measurements. The measurements from the morning (0000-1200 MST) of 17 April are highlighted in red. The dotted lines in Figs 13a and 13d show the emission ratios derived by *Gilman et al.* (2013) who applied multivariate regression analysis (MVR) to an extensive suite of VOC measurements made at the Boulder Atmospheric Observatory (BAO, cf. Fig. 10) during the NACHTT (Nitrogen, Aerosol Composition, and Halogens on a Tall Tower) campaign (18 February – 7 March 7 2011) (Brown et al., 2013). The dashed lines show the emission ratios from *Pollack et al.* (2021) who analyzed winter (December-February; DJF) and spring (March-May; MAM) measurements made at the Boulder Reservoir (BOUR) between 2017 and 2019 using both MVR and positive matrix factorization analysis (PMF) (Pollack et al., 2021). As was seen in those studies, Figs. 13a and 13b show that *n*-butane was highly correlated ($R^2=0.87$) with propane, but uncorrelated ($R^2=0.04$) with ethyne, while Figs. 13c and 13d show that ethene was uncorrelated ($R^2=0.04$) with propane, but highly correlated ($R^2=0.92$) with ethyne as seen in the earlier studies. The slope of the ODR fit (0.399±0.006) in Fig. 13a is smaller than that seen at the BAO nearly 10 years earlier, but comparable to that found in the more recent BOUR winter measurements. The slope in Fig. 13d is similar to that found in both of the earlier studies.

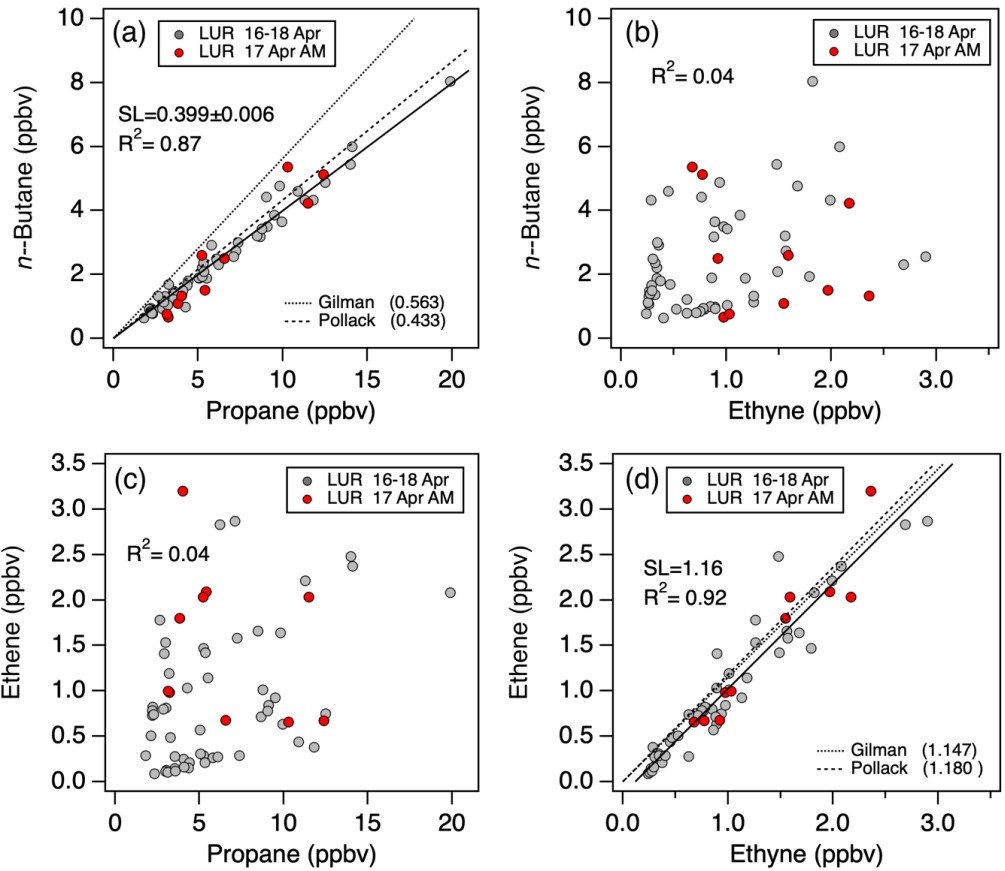

**Figure 13: Scatter plots comparing the LUR *n*-butane and ethene measurements to the propane (left) and ethyne (right) concentrations. The filled gray circles show all of the measurements made from April 16-18. The solid lines show the ODR regression fits to these measurements and the dotted lines the emission ratios from *Gilman* et al. (2013). The dashed lines show the winter (DJF) emission ratios from *Pollack* et al. (2022).**

The DSRC measurements in Figs. 11 and 12 appear both quantitatively and qualitatively different from the LUR measurements with most of the time series showing a nearly exponential buildup ($t \approx 8$ hours) on 17 April consistent with the transport of co-mingled traffic and O&NG emissions from the WGF and I-25 implied by the HYSPLIT back trajectories in Fig. 10. This co-mingling is shown by Fig. 14, which is similar to Fig. 13, but shows the DSRC measurements. The filled red circles highlight the measurements from the afternoon and evening (1200-2100 MST) of 17 April when the highest $O_3$ was measured. The solid red lines show the ODR fits to these measurements, and the solid, dotted, and dashed black lines are the same as in Fig. 13. In contrast to Figs. 13b and 13c, Figs. 14b and 14c show significant correlations between *n*-butane and ethyne (Fig. 14b), and between ethene and propane (Fig. 14c) at the DSRC, particularly on the afternoon and evening of the 17[th] (red points).



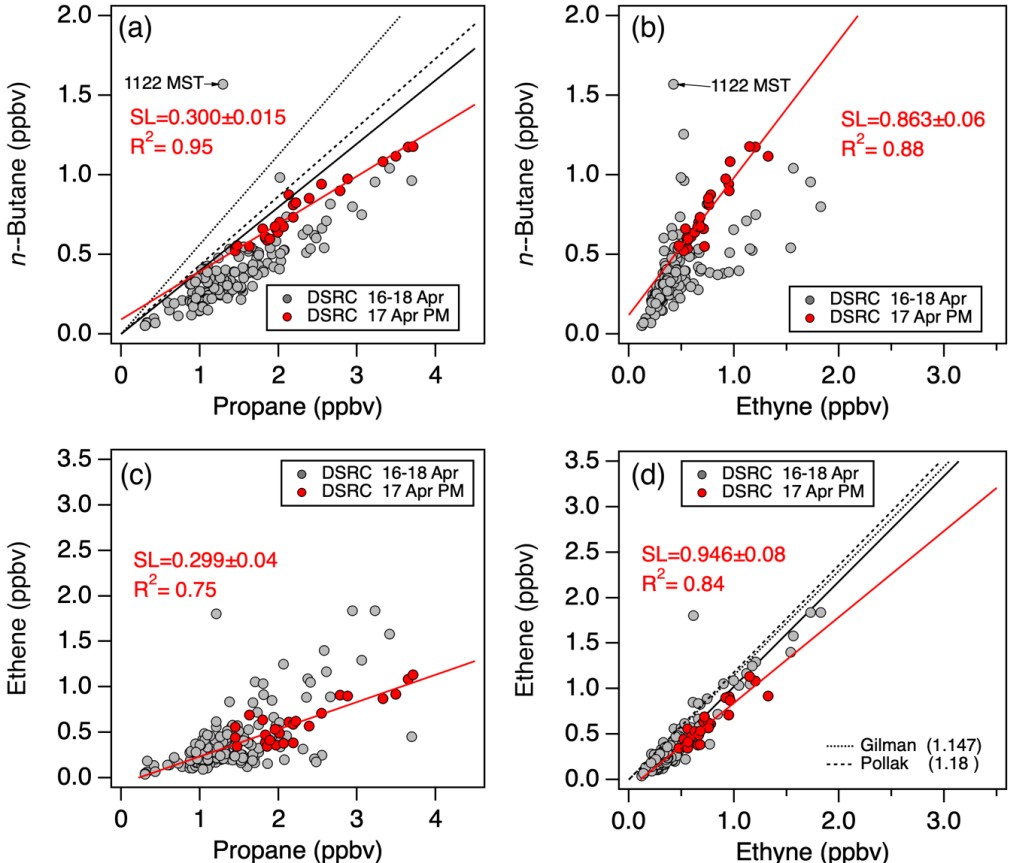

**Figure 14: Same as Fig. 13, but for the DSRC with the afternoon (MST) measurements from the 17th highlighted. Note the different scales. The solid red lines are the orthogonal distance regression (ODR) fits to the afternoon and evening (1200-2100 MST) measurements from the 17th, which are plotted in red. The solid, dashed, and dotted lines are from Figure 13.**

470

The relative contributions of motor vehicles and O&NG sources to the VOC burden can also be estimated from the relative abundances of the pentane isomers (Gilman et al., 2013). Raw natural gas samples collected near O&NG facilities in the WGF have a mean *i*-pentane/*n*-pentane ratio of 0.86±0.02 (Lt Environmental, 2007), but gasoline is

475 enriched in *i*-pentane compared to raw natural gas (Gentner et al., 2009) and the *i*-pentane/*n*-pentane ratio in air samples from heavily trafficked areas typically lie between 2 and 3 (Rossabi and Helmig, 2018). Figs. 15a and 15b, which are similar to Figure 2 of *Gilman et al.* (2013), compare the isomeric pentane measurements from the LUR and DSRC, respectively, with the mean values from previous studies. The solid lines correspond to the WGF raw natural gas ratio of 0.86 and the dotted lines the ratio of 2.41±0.02 derived from measurements made in Pasadena, CA during

480 the CalNex campaign (June-July 2010) (Borbon et al., 2013).



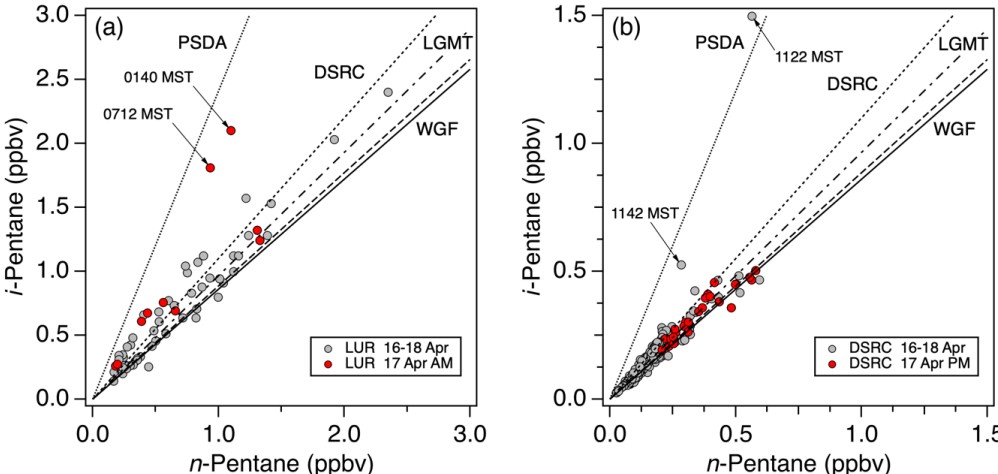

**Figure 15: (a) Scatter plots comparing the *i*-pentane and *n*-pentane measurements from the (a) LUR and (b) DSRC. Note the different scales. The solid line shows the mean slope (0.86) derived from WGF natural gas samples. The long- and short-dashed lines show relationships derived from previous measurements at the BAO, DSRC, and Pasadena, CA (PSDA) (*Gilman et al.*, 2013). The dot-dash line is from measurements in the Erie/Longmont area by *Thompson et al.* (2015).**

The other lines are from previous studies in the Boulder-Longmont area and range from the long dash line corresponding to the values of 0.885±0.002 and 0.89±0.01 derived from the winter measurements at the BAO (Gilman et al., 2013) and Boulder Reservoir (Pollack et al., 2021), to the short dash line (1.10±0.05) derived from two days of measurements at the DSRC in September 2010. The dot-dash line shows that the mean ratio of 0.965 derived from measurements made at multiple sites in the Longmont area between March and June of 2013 lies between these values (Thompson et al., 2015). All of these ratios show a much larger O&NG influence in the Boulder-Longmont area as do nearly all of the measurements from the LUR and DSRC plotted in Fig. 15. The only measurements showing significant motor vehicle influences are the two points corresponding to the two large traffic peaks (0140 and 0712 MST) in the morning LUR measurements (cf. Figs. 11 and 12) and the spikes at 1122 and 1142 MST attributed to snow removal operations at the DSRC.

**6.2 OH Reactivity**

The production of $O_3$ in the boundary layer is driven primarily by photolysis of $NO_2$ formed in the reaction of peroxy radicals (ROO•) with NO. The peroxy radicals are formed through a sequence initiated by the reaction of VOCs with the OH• radical and the "total OH reactivity" is often used as a measure of the $O_3$ production potential (Gilman et al., 2013). This quantity is calculated as:

$$R_{OH+VOC} = \sum(k_{OH+VOC} \times [VOC]) \qquad (1),$$

where $k_{OH+VOC}$ is the temperature and pressure dependent rate constant for the reaction with OH (Atkinson, 1986, 2003)  The contributions from $CH_4$ are not included since these were relatively constant. Fig. 16a plots the total OH reactivity calculated using the LUR measurements and a mean temperature of 278 K (5°C). The total reactivity ranged from 0.12 s$^{-1}$ on the afternoon of the 14$^{th}$ when the station was exposed to free tropospheric air, to 7.6 s$^{-1}$ when the



winds subsided later that night. Note that these totals do not include the mean contribution of 0.27 s$^{-1}$ calculated for CH$_4$, which varied little from day to day. The total reactivity is also resolved into the contributions from O&NG, urban, and biogenic sources, where we assume that O&NG operations were the primary source of C$_2$-C$_8$ alkanes and C$_5$-C$_8$ cycloalkanes, urban activities (motor vehicles, residential and commercial heating, power generation, and industrial activities) the primary sources of ethyne, C$_6$-C$_8$ aromatics, and alkenes, and biogenic emissions which are the primary

source of isoprene and pinene. The latter, which are important during summer, were negligible during this study. The concentrations of ethane, which was the most abundant VOC in both the LUR and DSRC measurements and is the second largest component of raw natural gas after methane, but also a major component of tailpipe emissions (Gentner et al., 2009), were resolved into the O&NG and urban contributions using the relationships from (Gilman et al., 2013) and the measured propane and ethyne.


Fig. 16a shows that O&NG emissions made the largest contributions to the total OH reactivity with the much higher alkane concentrations more than compensating for the greater reactivity of the alkenes emitted by motor vehicles. Previous studies in the area (Gilman et al., 2013; Swarthout et al., 2013; Abeleira et al., 2017) reached similar conclusions. The highest VOC concentrations were usually measured at night and Fig. 16b plots the reactivities scaled

by the relative LUR solar flux to emphasize the differences in photochemical activity. This shows that the total reactivity of 1.7 s$^{-1}$ on the afternoon of the 17$^{th}$ was more than an order of magnitude larger than the 0.12 s$^{-1}$ on the afternoon of the 14$^{th}$. This is about half the mean value of 3 s$^{-1}$ estimated from the wintertime measurements at the BAO (Gilman et al., 2013; Swarthout et al., 2013), Fig. 16c is similar to Fig. 16b, but shows the contributions of the individual VOCs to the O&NG total. The largest contributions were from propane, *n*-butane, and *n*-pentane, but our

estimates do not include the contributions of formaldehyde, acetaldehyde, or other OVOCs (Section 6.3) that were not measured at LUR, but accounted for approximately 25% of the total reactivity in those studies. Ethanol (C$_2$H$_5$OH), which comprises about 10% of Colorado winter gasoline (De Gouw et al., 2012), was the second most abundant VOC at the DSRC, but has low ozone forming potential.


**Figure 16:** Calculated OH reactivities for the non-methane VOCs measured at Union Reservoir. **(a)** Total reactivity resolved into the contributions from O&NG, urban, and biogenic sources. **(b)** Same as (a), but normalized by the peak solar flux on 17 April. **(c)** Same as (b), but showing the contributions of the individual alkanes to the O&NG total.





### 6.3 Chemical Box Model

The shallow boundary layer and weak winds on 17 April created the ideal conditions for the application of a chemical box model (Edwards et al., 2014). Here, we use the "Framework for 0-D Atmospheric Modeling (F0AM)" box model (Wolfe et al., 2016) with the more extensive suite of VOCs, OVOCs, and other compounds (Supplement) measured

at the DSRC during the spring and summer of 2020 as part of COVID-AQS (Peischl et al., 2023). Our application closely follows that of *Rickly et al*. (Rickly et al., 2023) who used the F0AM model with the same measurement suite to investigate the impact of wildfire smoke on $O_3$ production in Boulder during the late summer and early fall of 2020. As in that study, the meteorological inputs including pressure, temperature, and relative humidity, were obtained either from local measurements or WRF-Chem, and the boundary layer height (BLH) and low-level winds were taken from

the Doppler lidar measurements at the DSRC. We use the Niwot Ridge measurements (cf. Fig. 4) to estimate the baseline $O_3$, and the NOAA GML Solar Position Calculator (https://gml.noaa.gov/grad/solcalc/azel.html, last access 24 June 2024) to calculate the solar zenith angle. Fig. 17 compares the $O_3$ mixing ratios calculated by the box model with the DSRC measurements. The model did an excellent job of reproducing the April measurements with a net photochemical production of 3 ppbv of $O_3$ on 14 April and 22 ppbv on 17 April.


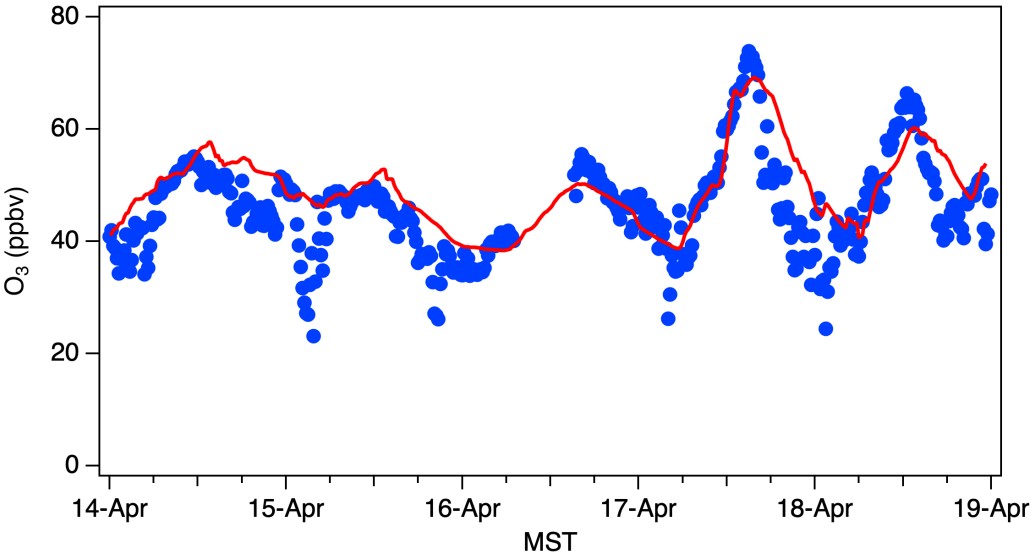

**Figure 17: Measured (filled blue circles) and simulated (solid red line) $O_3$ at the DSRC.**

The box model simulations also showed that the $O_3$ production on both days was driven by photolysis of formaldehyde,
acetaldehyde, and other oxygenated compounds that acted as radical amplifiers. This was also found to be the case in the Uinta Basin (Edwards et al., 2014) and a recent analysis (Dix et al., 2023) of measurements from the Shale Oil and Natural Gas Nexus (SONGNEX) aircraft campaign concluded that 96±3% of the above background formaldehyde in the Uinta Basin, Denver-Julesburg Basin, and 7 other O&NG fields sampled during March and April 2015 was a



secondary product formed by photooxidation of the primary VOC emissions. The more populated parts of the
DM/NFR have other potential sources of formaldehyde, however, including direct emissions from motor vehicles,
power plants, residential/commercial heating systems, and other combustion sources (Bastien et al., 2019; Green et
al., 2021), but Fig. 18 shows that the $CH_2O$ mixing ratios measured at the DSRC on 17 April were much more strongly
correlated with $n$-$C_4H_{10}$ than with $C_2H_2$ and $C_2H_4$, suggesting that most of this formaldehyde was also derived from
O&NG sources. The weaker correlations with $C_2H_2$ and $C_2H_4$ in Figs. 18b and 18c can also be explained by the
mingling of urban and O&NG emissions as was seen in Fig. 14. Fig. 18a also shows that $CH_2O$ and $n$-$C_4H_{10}$ were
essentially uncorrelated in the measurements from 14 April-16, confirming that the formaldehyde was not a primary
emission.

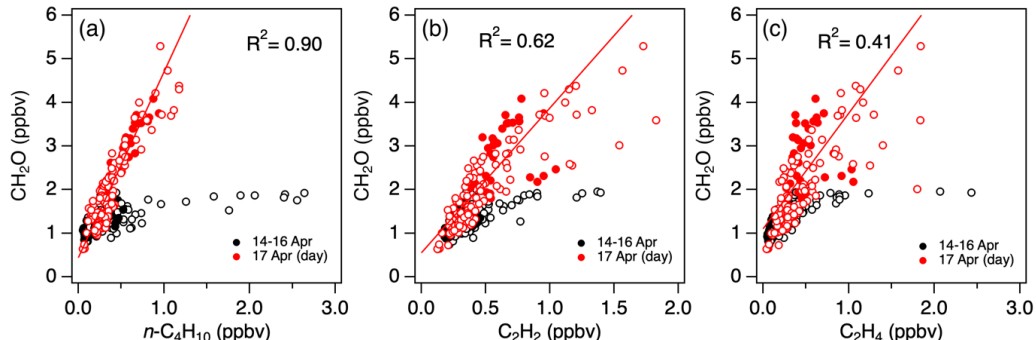

**Figure 18: Scatter plots comparing the $CH_2O$ mixing ratios measured at the DSRC on 14-16 April (black) and 17-18 April
(red) with the co-incident measurements of (a) $n$-butane, (b) ethyne, and (c) ethene. The ODR fits and correlation coefficients
are for the 17 April daytime measurements (0700-1700 MST) represented by the filled red circles. The 30-s PTR-MS
measurements have been interpolated to match the longer integration time of the whole air sampler.**

The 17 April 2020 high $O_3$ episode occurred early in the COVID-19 lockdown when motor vehicle traffic and the
associated $NO_x$ emissions were greatly reduced (Harkins et al., 2021). $NO_x$ sensitivity tests (Fig. 19) show that there
was still more than enough $NO_x$ to push $O_3$ production into the $NO_x$-saturated regime on both days, however, in
contrast to the $NO_x$-sensitive regime typical of summer (Mcduffie et al., 2016; Rickly et al., 2023). This difference
may be due in part to the absence of isoprene and other highly reactive biogenic VOCs during late winter and early
spring. The traffic counts along the stretch of I-25 nearest the LUR monitor and US 36, which connects Denver with
Boulder and passes within 2 km of the DSRC, in April of 2020 were down to about 35 and 45% of the April average
for 2017-2019 (https://dtdapps.coloradodot.info/otis, last access 24 June 2024), and if we assume that the $NO_x$
emissions decreased by a similar amount, the sensitivity curves in Fig. 19 suggest that the $O_3$ production on the 17[th]
would have been about 30% smaller (open squares) than was measured in 2020 (filled squares) without the COVID-
19 quarantine and the NAAQS exceedances probably would not have occurred.



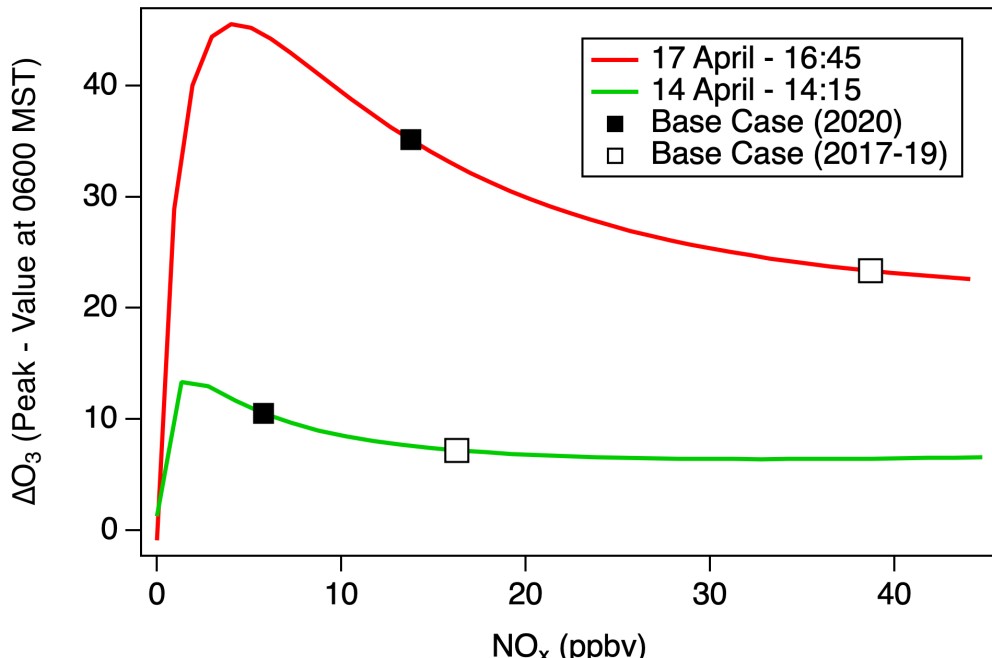

**Figure 19: NO$_x$ sensitivity curves from the chemical box model calculations for 14 and 17 April. The filled squares show the O$_3$ production calculated from the 1000 to 1600 MST mean NO$_x$. The open squares show the production expected if the NO$_x$ were doubled to reflect 2019 traffic volumes.**


## 7. Summary and Conclusions

The high O$_3$ event that occurred along the Northern Front Range of Colorado on 17 April 2020 was in many ways
similar to the wintertime O$_3$ events endemic to the Uinta Basin of northeastern Utah and Upper Green River Basin of
southwestern Wyoming. Like those events, the photochemical production and accumulation of O$_3$, PM$_{2.5}$, and other
pollutants was caused by the buildup of NO$_x$ and VOCs from anthropogenic sources within an extremely shallow
boundary layer combined with clear skies and snow cover to increase the surface albedo and hence the effective solar
flux. Like those events, the reactive VOCs appear to have originated primarily from O&NG activities, but in this case
the NO$_x$ appears to have come primarily from motor vehicles rather than the combustion sources associated with the
O&NG industry (*e.g.*, drilling rigs, compressors, pumping stations, etc.). If so, the event appears to have been
exacerbated by the decrease in NO$_x$ that followed the reduction in motor vehicle traffic during the COVID-19
quarantine. This implies that similar winter events might become more common in the Colorado Northern Front Range
as emissions controls reduce NO$_x$ levels through the transition region of O$_3$-NO$_x$ sensitivity, even though recent box-
model analyses suggest that such NO$_x$ reductions may mitigate summertime O$_3$ in the region. A search through the
previous 10 years of CDPHE measurements found no obvious examples of similar episodes, but NO$_x$ has been steadily
decreasing in the DM/NFR over the last two decades (Langford et al., 2023) resulting in higher VOC/NOx ratios and
possibly leading to more of these early season ozone events in future. Indeed, another high winter O$_3$ event occurred



less than one year later on 19-20 March 2021 (Caputi et al., submitted to J. Geophys. Res.). Unfortunately, the ozone
       lidar and COVID-AQS *in-situ* measurements used in the present study were not available for an analysis comparable
       to that described here.

       *Data availability.* The regulatory $O_3$ measurements used in this study are available at https://www.epa.gov/outdoor-
air-quality-data (last access 24 June 2024). The A.I.R. monitoring data have been submitted to EBAS
       (https://ebas.nilu.no/). The NOAA COVID-AQS data are available at
       https://csl.noaa.gov/groups/csl7/measurements/2020covid-aqs/ (last access 24 June 2024) and the TOPAZ lidar data
       at https://tolnet.larc.nasa.gov/ (last access 24 June 2024).

       *Supplement.* The supplement related to this article is available online at:

*Author contributions.* RJA, WAB, SB, BJM, CJS, and SPS acquired, analyzed, and archived the NOAA CSL
       TOPAZ and Doppler lidar data. KCA, SSB, MMC, JG, GIG, ADL, AMM. JP, PSR, AWR, and CW acquired,
       analyzed, and archived the NOAA CSL COVID-AQS measurements. PDC, BJJ, AMB, and IP acquired, analyzed,
       and archived the NOAA GML ozonesonde and surface measurements. DH provided the Boulder A.I.R.
       measurements. MMC, SB, and PSR conducted the box model simulations, RK and GP the NCAR WFR-Chem
model analyses, and KEK the GEOS-CF model analyses. AOL assimilated the data and wrote the manuscript with
       the coauthors feedback.

       *Competing interests.* The authors declare that they have no conflict of interest.

       *Disclaimer.* The scientific results and conclusions, as well as any views or opinions expressed herein, are those of the
       author(s) and do not necessarily reflect the views of NOAA or the Department of Commerce.

*Acknowledgements.* The authors would like to thank Catherine Burgdorf-Rasco for creating and maintaining the
       CSL data archive. We would also like to thank Scott Landes, Dan Welsh, and Erick Mattson of the Colorado
       Department of Public Health and the Environment for help with the monitoring data

       *Financial support.* The NOAA CSL TOPAZ lidar operations were supported in part by the NASA-sponsored
       Tropospheric Ozone Lidar Network (TOLNet). The monitoring of meteorological and chemical atmospheric data by
Boulder A.I.R. was funded by the City of Longmont and Boulder County. Resources supporting the GEOS-CF model
       simulations were provided by the NASA High-End Computing (HEC) Program through the NASA Center for Climate
       Simulation (NCCS) at Goddard Space Flight Center. The WRF-Chem simulations are based upon work supported by
       the NSF National Center for Atmospheric Research, which is a major facility sponsored by the NSF under Cooperative
       Agreement No. 1852977KCA, SB, MC, PDC, GIG, ADL, AM-B, BJM, JP, IP, PR, and CJS were supported in part by
NOAA cooperative agreements NA17OAR4320101 and NA22OAR4320151.



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
