# Peer review of "An Unusual Winter Ozone Event in Colorado"

_EGUsphere, 2024_

## Author Comment (AC1)

We would like to thank the reviewer for their comments which are parsed and reproduced here in boldface italics along with our responses.

***The analysis of the ozone exceedance event on April 17, 2020, in the Boulder-Fort Collins region contains several intriguing elements. While the individual aspects of the analysis appear sound, the combination of them appears to be problematic. I suggest the authors focus on the most compelling aspect and conduct a more in-depth analysis. This focused approach would likely yield more robust and meaningful conclusions.***

We do not understand what the reviewer means when they suggest we "…focus on the most compelling aspect and conduct a more in-depth analysis." What do they consider most compelling? We would argue that all components of our already "in-depth" analysis are essential to understanding this episode.

***This April event differs significantly from previously reported winter ozone episodes. While winter events have recorded ozone levels up to 150 ppb, this springtime occurrence peaked below 80 ppbv. A key factor in winter high ozone episodes is a shallow boundary layer. It would be beneficial to compare its boundary layer height (BLH) with those of winter episodes. This comparison could help explain whether the typically higher springtime BLH contributes to the lower observed ozone levels. A more appropriate paper title would be "An Unusual Winter-like Ozone Event in Colorado."***

The reviewer is correct that the episode described here differs from those that occur in O&NG basins of northeastern Utah and southwestern Wyoming in several important ways. The primary reason for the lower peak $O_3$ levels in the present case was not the *depth* of the boundary layer, but rather its *persistence*. The basin-like terrain of the Utah and Wyoming O&NG fields supports the formation of stable shallow cold air pools that can persist for days or even weeks until they are disrupted by a frontal passage. This allows $O_3$ to accumulate near the surface over time, and in the example described by *Edwards et al.* (2014), the $O_3$ concentrations increased from ≈80 ppbv, on Day 1, which is similar to that observed in the short-lived event described here, to more than 120 ppbv on Day 5.

***The assertion that "statically stable lower stratospheric air suppressed the growth of the convective boundary layer" requires further substantiation. Figure 8 indicates a BLH of 200m, which is inconsistent with a convective boundary layer.***

The term "convective boundary layer" refers to the region above the surface where the air is well mixed, whether by free convection (thermals) or forced convection (mechanical turbulence). The BL depth of 200 m is entirely consistent with the latter.

***If stratospheric air indeed descended to such a low altitude (200 m), one would expect significantly higher ozone concentrations than those shown in Figure 7.***

*Not necessarily if there was limited mixing at the top of the convective boundary layer as we show to be the case in Fig. 8.*

*A more obvious explanation for the shallow BLH could be the low surface temperature shown in Figure 2. Introducing a complex mechanism involving stratospheric air seems unnecessary without stronger supporting evidence. The authors should either provide more robust data to support this claim or consider alternative explanations that align more closely with the observed conditions.*

if the snow cover and low temperatures alone had been sufficient to create the 200 m deep boundary layer on 17 April, then we would have expected to see a very similar boundary layer depth on 14 April when the temperatures and snow cover were similar. Instead, we measured a significantly larger depth of 630 m. We contend that the capping of the boundary layer by the stratospheric intrusion was a key factor in the evolution of the 17 April event and that this is thoroughly documented by the lidar and ozonesonde measurements presented in the manuscript. There is nothing particularly "complex" about the mechanism we describe here. Stratospheric intrusions form above the western U.S. during the passage of most cyclonic systems in late winter and spring, and many examples have been documented by the TOPAZ lidar in Boulder. Most of these intrusions descend only into the middle free troposphere, however, and very few penetrate all the way to the surface (see *Langford et al.* 2009 for an example of the latter).

*Please include "baseline" ozone concentrations at the NWR site in Figure 17. It appears that model simulated ozone concentrations at the DSRC are not much higher than NWR on April 17 and 18. Is this modeling result appropriate for diagnosing the high ozone event shown in Figure 4?*

We have attached a modified version of the figure that includes both the "baseline" measurements from NWR and the highest $O_3$ measurements from BOS. The reviewer is correct in that the simulated (and measured) $O_3$ concentrations at the DSRC are only about 8 ppbv higher than the "baseline" concentrations. This may not seem like much, but it is much greater than the difference of ≈1 ppbv measured on 14 April when the temperatures, snow cover, and insolation were similar.

 *Several additional questions arise from the F0AM model analysis:*

1. *How does the significantly lower VOC concentration at DSRC compared to LUR and BOUR (Figure 12) impact the model's sensitivity to VOCs, especially given that NOx levels are comparable across sites (Figure 11)?*
2. *Would F0AM simulations using NOx and VOC concentrations from LUR and BOUR reproduce the observed ozone concentrations at those sites?*
3. *Can the variations in NOx and VOC concentrations explain the lower ozone levels at LUR and higher levels at BOUR relative to DSRC?*
4. *If these concentration differences do not fully account for the observed ozone variations, what is the justification for using the model results to predict an increased frequency of high ozone events in the future?*

We have limited the F0AM model analysis in our manuscript to the DSRC because there were no measurements of the boundary layer depths at the BOUR or LUR and the limited measurement

suites at these monitoring stations did not include formaldehyde, acetaldehyde, or any of the other oxygenated compounds that were found to dominate the $O_3$ production.

*The attribution of VOCs to oil and natural gas (O&NG) sources and NOx to motor vehicles in this event raises an important question: Why aren't similar ozone exceedance events observed more frequently in nearby regions with comparable source combinations? The co-location of O&NG fields and major highways like I-25/I-70 is not unique to this area. This scenario suggests that additional factors beyond the presence of these emission sources must be at play.*

Indeed. We would argue that the additional factor in this case was a stratospheric intrusion.

References

Langford, A. O., Aikin, K. C., Eubank, C. S., and Williams, E. J.: Stratospheric contribution to high surface ozone in Colorado during springtime, Geophys. Res. Lett., 36, doi:10.1029/2009GL038367., 2009).

Edwards, P. M., et al.: High winter ozone pollution from carbonyl photolysis in an oil and gas basin, Nature, 514, 351-+, 10.1038/nature13767, 2014.

---

## Author Comment (AC2)

**RC2**: ['Comment on egusphere-2024-1938'](), Anonymous Referee #3, 25 Sep 2024

We would like to thank the reviewer for their comments which are reproduced here in boldface italics along with our responses in plain text.

*General comments:*

*The manuscript describes an unusual winter ozone event in Colorado during a day with high photochemical activity, which was enhanced by a snow-covered surface. The analysis uses a combination of surface measurements, lidar, ozone sondes, HYSPLIT back trajectories, and models to show the origin of the enhanced O3 observed. While the paper is structured differently than most other measurement papers (no separation between methods and results), it guides the reader through the meteorological conditions needed for such an event, observations of O3 at multiple sites around Boulder, Colorado, and results from a chemical box model for one of the sites, where a large suite of VOC data was measured. It would have been interesting to see chemical box modelling for multiple sites to see if the O3 measurements at the highest exceedance site could be reproduced, however, without having measured VOCs at this location during this event it seems unrealistic.*

*My recommendation is to publish the manuscript after taking into account the minor corrections/comments below.*

*Minor comments:*

*Line 156-159: In the text you write that "The measurements from the high elevation (>2.5 km above mean sea level, a.s.l.) "baseline" monitors at Blackhawk (BHWK), NWR, and Rocky Mountain National Park (RMNP) are plotted in black with the measurements from those monitors that recorded higher (lower) MDA8 O3 on the 17th plotted in red (blue).", which sounds like all monitors that measured higher (lower) MDA8 O3 on the 17th than the baseline monitors are plotted in red (blue). However, in the plot it is only the highest (lowest) measurement of MDA8 O3 on the 17th plotted in red (blue), which is described in the following sentence. All other monitors are plotted in grey.*

The text has been revised to eliminate the confusion.

*Figure 4: The horizontal dashed lines indicating the 2015 NAAQS should be the same colour in both panels.*

Fixed.

*Figure 6: It is difficult to read the potential temperatures in the red shaded areas because of the lines being very close to each other (and the numbers therefore overlapping) and due to the text disappearing in the dark red shading.*

The stratospheric potential temperature values are not important for this study, but we have enlarged the figure to improve the readability.

*Figure 9: In the figure text you write that the dashed lines are the 200 m PBL height, but in panel (c) the line is not at 200 m. From above in the text, it sounded like the PBL height was higher on the 14th than on the 17th, so the text should either specify both heights individually or just say that it is the PBL height at the time of the hysplit trajectory initiation.*

The figure caption has been amended.

*Figure 10: In the figure text about panel (a), the dashed lines for the hysplit trajectories from the BOS station are not mentioned anywhere. They are mentioned in the text, but the difference between solid and dashed lines is not.*

The figure caption has been amended.

*Line 354-358: It would be good to mention that the altitude profiles mentioned here are shown in Fig 10c.*

Done.

*Figure 13: For extra clarity the text should be "Scatter plots comparing the LUR n-butane (top) and ethene (bottom) measurements to the propane (left) and ethyne (right) concentrations." The year of the Pollack reference is written as 2021 in the text (line 434-440), but 2022 in the figure text. And a description of which hours the red data covers could also be explained in the figure text.*

Done.

*Line 458: You write that the co-mingling of traffic and O&NG emissions is implied by the back trajectories in figure 10, however, only one of the DSRC trajectories in figure 10b crosses the I-25 mentioned in the text. Could there be another traffic emission source than the I-25 involved? Or is it better to describe it as mixing of urban and O&NG emissions as you do in line 570 when referring to figure 14?*

We have revised the sentence to read "…co-mingled traffic and O&NG emissions from *Longmont*, I-25, and the WGF implied…"

*Line 478-480: From the text it is implied that the ratio of 2.41 is typical for traffic emissions of i-pentane/n-pentane, however, it would be good to add some context to the sentence such as what the focus of the CalNex campaign focused on instead of implying that it measured traffic emissions.*

CalNex was a major field campaign with both airborne and ground-based elements that involved dozens of researchers. A comprehensive description of the multiple objectives is not really relevant here, but we have added a reference to the overview paper by Ryerson et al. for those who are interested.

*Figure 15: In the text it is mentioned that the dotted line represents the Pasadena measurements, however, in the figure text the dotted line is not mentioned despite Pasadena being mentioned: "The long- and short-dashed lines show relationships derived from previous measurements at the BAO, DSRC, and Pasadena, CA (PSDA) (Gilman et al., 2013)."*

Fixed.

*Figure 13, 14, and 15: Figure 13 show that the LUR site predominately measures O&NG emissions based on the good correlation between n-butane/propane and ethene/ethyne and Figure 14 shows that the DSRC site measures a combination of O&NG and traffic emissions during the afternoon/evening of the 17th based on good correlation between n-butane/propane and ethene/ethyne as well as n-butane/ethyne and ethene/propane. However, in figure 15 the correlation between i-pentane/n-pentane at the DSRC site looks closer to the O&NG ratio than the LUR site. How do you explain that?*

The O&NG influence was much smaller at the DSRC, but the COVID shutdown and recent snowfall impacted the local and commuter traffic in Boulder more than the commercial traffic on I-25 so that the *relative* contribution of the O&NG sources was larger. The reviewer may have overlooked the different scales in the two plots so we have added the phrase "Note the different scales on the two plots" to the figure caption.

*Line 524-525: My understanding of the scaled OH reactivity mentioned is that it is used to show the O3 producing potential of the measured air is higher on April 17th since O3 production is dependent on sunlight. Could the purpose of scaling the OH reactivity be added to the text if that is correct? If it is not correct, then some additional explanation is needed to understand why you scale the OH reactivity to the solar radiation.*

The solar flux was actually very similar on the 14th and 17th, but we have scaled the solar flux to emphasize the greater photochemical production on the clear days (14th and 17th) compared to the cloudy days (15th, 16th, and 18th). We have revised the text to make this clearer.

*Figure 16: Would it be clearer if the different OH reactivity contributions were stacked on top of each other so you can see the total reactivity from the plot as well?*

The total reactivity is already plotted (heavy black lines) in panels (a) and (b), and panel (c) shows the total O&NG contribution in red.

*Technical corrections:*

*Line 152: The ")" after DM/NFR should be removed.*

    Fixed.

*Line 279: Insert comma: "… potential temperature (theta), relative 280 humidity (RH)…"*
    Fixed.

*Line 327: Insert "the": "Even the trajectory launched from the RFN monitor…"*
    Fixed.

*Line 364: Delete one of the "passing" in the sentence.*
    Fixed.

*Line 546: Change format of reference from "… that of Rickly et al. (Rickly et al., 2023)…" to "… that of Rickly et al. (2023)…"*
    Fixed.

*Line 612: NOx should be $NO_x$*
    Fixed.

*Supplementary information:*

*Page 4, line 2: "in-situ" should be "in-situ"*
    Fixed.

*Page 4, line 6: The Rickly reference is not in the reference list and assuming it is the same reference mentioned in the manuscript, the year should be 2023*
    Fixed.

**Citation**: https://doi.org/10.5194/egusphere-2024-1938-RC2